# Molecular basis of regio- and stereo-specificity in biosynthesis of bacterial heterodimeric diketopiperazines

Chenghai Sun [1,2,4], Zhenyao Luo[3,4], Wenlu Zhang[2], Wenya Tian[1,2], Haidong Peng[2], Zhi Lin[1], Zixin Deng[1,2], Bostjan Kobe [3✉], Xinying Jia [3✉] & Xudong Qu [1,2✉]

Bacterial heterodimeric tryptophan-containing diketopiperazines (HTDKPs) are a growing family of bioactive natural products. They are challenging to prepare by chemical routes due to the polycyclic and densely functionalized backbone. Through functional characterization and investigation, we herein identify a family of three related HTDKP-forming cytochrome P450s (NasbB, Nas$_{S1868}$ and Nas$_{F5053}$) and reveal four critical residues (Qln65, Ala86, Ser284 and Val288) that control their regio- and stereo-selectivity to generate diverse dimeric DKP frameworks. Engineering these residues can alter the specificities of the enzymes to produce diverse frameworks. Determining the crystal structures (1.70–1.47 Å) of Nas$_{F5053}$ (ligand-free and substrate-bound Nas$_{F5053}$ and its Q65I-A86G and S284A-V288A mutants) and molecular dynamics simulation finally elucidate the specificity-conferring mechanism of these residues. Our results provide a clear molecular and mechanistic basis into this family of HTDKP-forming P450s, laying a solid foundation for rapid access to the molecular diversity of HTDKP frameworks through rational engineering of the P450s.

[1] State Key Laboratory of Microbial Metabolism and School of Life Sciences and Biotechnology, Shanghai Jiao Tong University, 200240 Shanghai, China. [2] Key Laboratory of Combinatorial Biosynthesis and Drug Discovery, Ministry of Education, Wuhan University School of Pharmaceutical Sciences, 430071 Wuhan, China. [3] School of Chemistry and Molecular Biosciences, Institute for Molecular Bioscience and Australian Infectious Diseases Research Centre, The University of Queensland, St. Lucia, QLD 4072, Australia. [4] These authors contributed equally: Chenghai Sun, Zhenyao Luo. ✉email: b.kobe@uq.edu.au; x.jia1@uq.edu.au; quxd19@sjtu.edu.cn

atural products derived from tryptophan-containing diketopiperazine (TDKP) comprise a large class of secondary metabolites[1-3]. Among them, heterodimeric tryptophan-containing diketopiperazines (HTDKPs) are particularly attractive for their unique structural architecture and fascinating bioactive properties, ranging from anticancer, anti-plasmodial, anti-HIV, and neuroprotective activities[1-6]. TDKPs are primarily produced by fungal systems, in which two pyrroloindoline units are predominantly fused together by a C3-C3' bond[3,7,8]. Bacterially-sourced HTDKPs are much rarer, but their structural architectures are more versatile[4-6,9-11]. Based on their connectivities and stereochemistry, the dimeric DKP frameworks can be classified into five different types (Fig. 1): (I) C3-C7', 2R-3S (e.g., naseseazine B or NAS-B[4,9]); (II) C3-C7', 2S-3R (e.g., NAS-3[6]). (III) C3-C6', 2S-3R (e.g., naseseazine C or NAS-C[5]); (IV) C3-C6', 2R-3S (e.g., iso-NAS-B[11]); and (V) N1-C7' (e.g., aspergilazine A or Asp-A[10]).

The regio-specificity and stereo-specificity in the densely functionalized frameworks, especially at the quaternary stereo-center at the C3 position, renders chemical synthesis of bacterial HTDKPs very challenging[9,12-14]. To develop efficient biocatalytic approaches, we recently investigated the biosynthesis of naseseazine C (NAS-C) and identified a key diketopiperazine (DKP) forming P450 enzyme (NascB)[6]. NascB catalyzes a radical cascade reaction to form intramolecular and intermolecular carbon–carbon bonds with both regio-specificity and stereo-specificity, which is very efficient in constructing the HTDKP frameworks and has been used to create 30 type I–IV NAS analogs employing different DKP substrates[6]. Very recently, Li and coworkers further identified two other P450 enzymes, AspB and NasB, which are responsible for the predominant formation of aspergilazine A (ASP-A) and NAS-B, respectively[11] (Supplementary Table 1). Unusually, HTDKP-forming P450s have relaxed regio-specificity and stereo-specificity and can generate

products with different frameworks, e.g., AspB is able to convert cyclo-L-Trp-L-Pro ($cW_L$-$P_L$) into NAS-C (type III) and iso-NAS-B (type IV) accompanying the major product ASP-A (type V)[11]. This property of co-generation of different types of HTDKPs suggests these P450s have a regulatory mechanism in controlling different regio-specificities and stereo-specificities and presents a great potential for further improving catalytic efficiency, altering specificity and even creating diverse frameworks by rational protein engineering. However, such endeavors are reliant on understanding the molecular basis of HTDKPs-producing P450-catalyzed reactions, which currently remains elusive.

To this end, we herein functionally characterize three HTDKP-forming P450s (NasbB, $Nas_{S1868}$, and $Nas_{F5053}$), structurally characterize $Nas_{F5053}$ and its mutants by X-ray crystallography, and explore the further catalytic potential of the fourth pertinent P450 (NascB). Our results reveal that four key residues (Q65, A86, S284, and V288, according to the $Nas_{F5053}$ numbering; PDB ID 6W0S) are crucial for controlling the combination of the different regio-specificities and stereo-specificities. Based on our structural characterization, molecular dynamics and mutagenesis-validation of the residues involved, we elucidate how the regio-configuration and stereo-configuration in forming bonds is finely tuned in these P450s.

## Results

**Identifying the P450s producing NAS-B and ASP-A.** Previously, we identified three distinct loci (locus-1, 2, and 3) in the *Streptomyces* CMB-MQ030, each of which contained genes encoding one cyclodipeptide synthase (CDPS) and one adjacent P450. The P450 NascB (CYP nomenclature: CYP1190B2) encoded in locus-1 was responsible for the biosynthesis of NAS-C[6]. To verify the functionality of locus-2, its P450 NasbB was expressed in the *Mycobacterium* system. In the presence of spinach flavodoxin (Fd) and flavodoxin reductase (FdR), as well as an NADPH

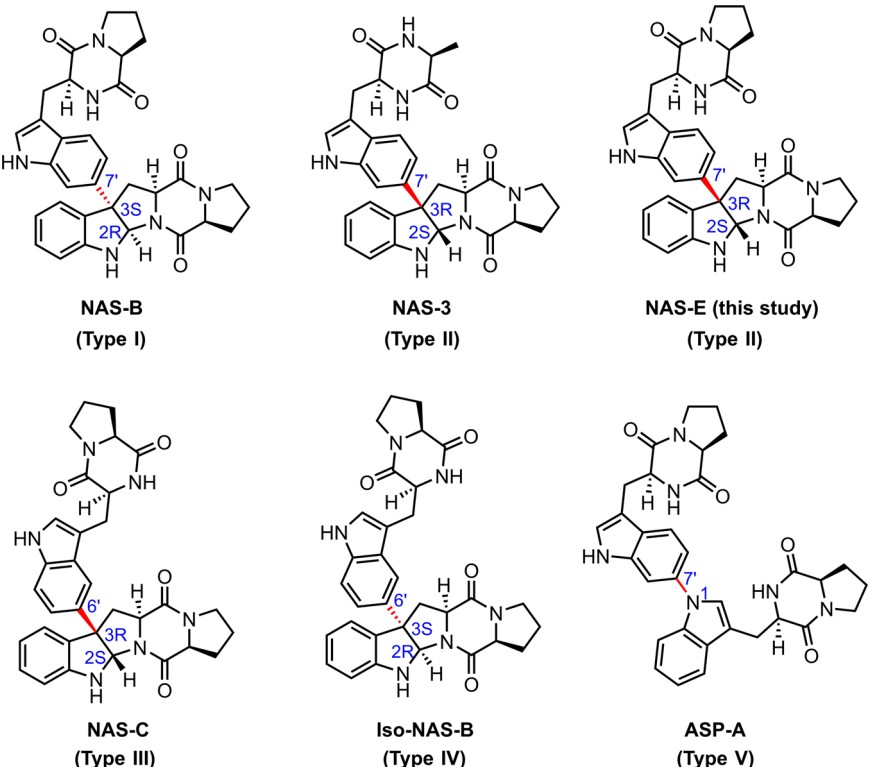

**Fig. 1 The structures of representative bacterial HTDKP natural products.** The bond connectivity of two DKP moieties (red) and the C2-C3 chirality (blue labels) are highlighted. NAS-E was produced in this study.

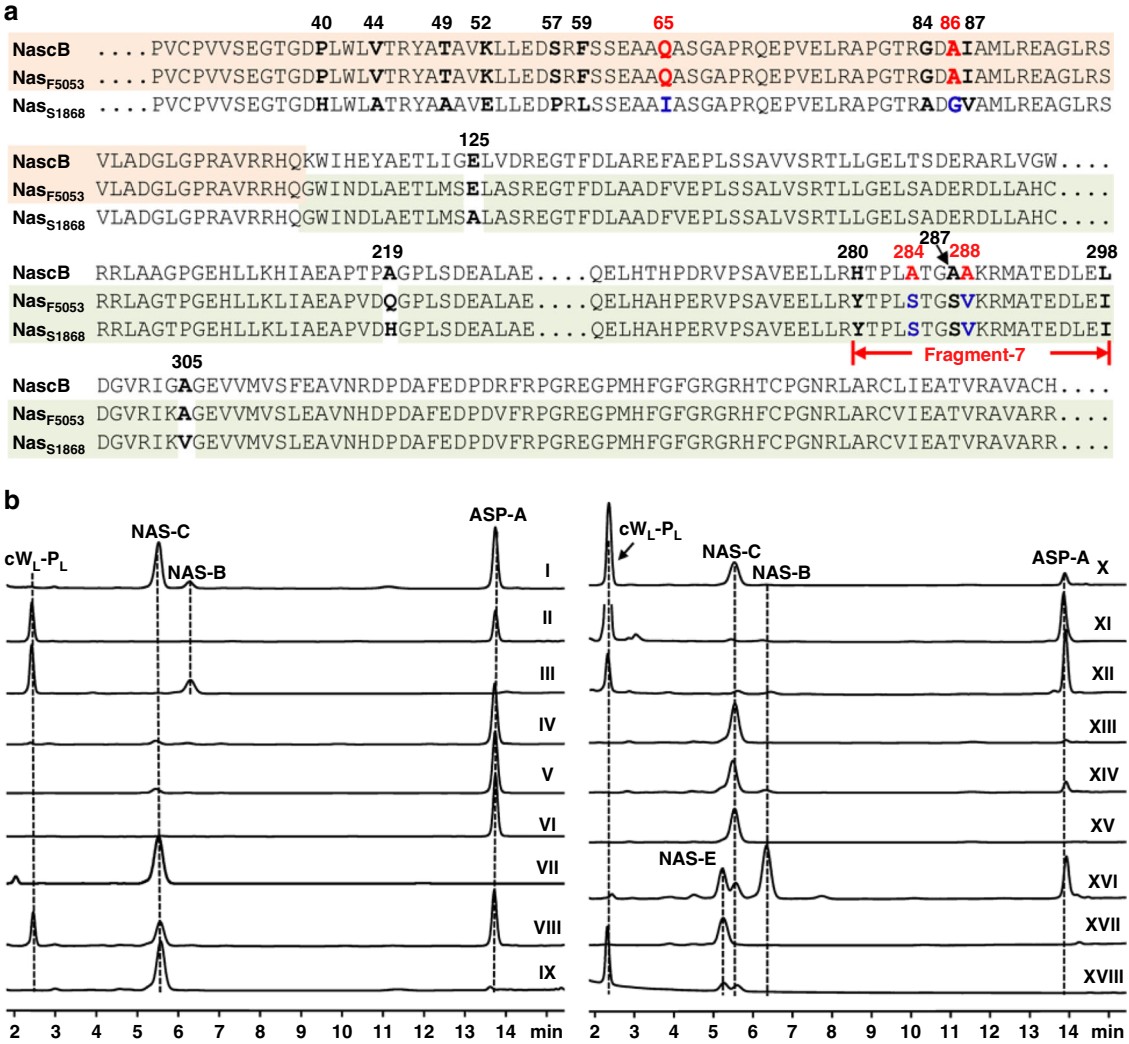

**Fig. 2 Deciphering and engineering the regio-specificity and stereo-specificity of HTDKP forming P450s. a** The sequence alignments of NascB, Nasc$_{F5053}$, and Nas$_{S1868}$. The identical N-terminal and C-terminal parts of NascB/NasF$_{5053}$ and Nas$_{F5053}$/Nas$_{S1868}$ are shaded in orange and light green, respectively. Residues less important are not shown and indicated by dashed lines. The four critical residues are bolded and highlighted by colors. **b** In vitro characterization of P450s and their mutants using cW$_L$-P$_L$ as substrate. (I) Nasc$_{F5053}$; (II) Nas$_{S1868}$; (III) NasbB; (IV) Nas$_{F5053}$-Q65I; (V) Nas$_{F5053}$-A86G; (VI) Nas$_{F5053}$-Q65I-A86G; (VII) NascB; (VIII) NascB-Q65I; (IX) NascB-A86G; (X) NascB-Q65I-A86G; (XI) NascB-S1868fragment-7; (XII) NascB-Q65I-A86G-A284S-A288V; (XIII) Nasc$_{F5053}$-S284A; (XIV) Nasc$_{F5053}$-V288A; (XV) Nasc$_{F5053}$-S284A-V288A; (XVI) Nasc$_{F5053}$-A86K-V288P; (XVII) NAS-E synthetic standard; (XVIII) Nasc$_{F5053}$-Q65P-A86W-S284C.

recycling system (NADP, glucose and glucose dehydrogenase), NasbB was confirmed to efficiently dimerize the cW$_L$-P$_L$ into NAS-B (Fig. 2b, trace III; NMR and HRMS data see Supplementary Fig. 1, Supplementary Fig. 2 and Supplementary Table 2). In the meantime, Li et al identified a homologous P450 (NasB, 96% identity to NasbB) from *Streptomyces* NRRL S-1868, also generating NAS-B[11].

As the P450 enzymes with highly similar sequences can produce different products, we were interested in establishing the relationship between enzyme sequences and the corresponding products. Although NascB and NasbB share 68% sequence identity, it is difficult to extract the key residues responsible for the difference in product formation. In order to identify more P450s which could potentially generate some other kinds of C3-aryl pyrroloindolines, we used simple sequence searches to find genes homologous to *nascB* or *nasbB*. We identified two previously uncharacterized P450 proteins: Nas$_{F5053}$ and Nas$_{S1868}$ from the *Streptomyces* strain sp. NRRL F-5053 and sp. NRRL S-1868, respectively. Soluble recombinant Nas$_{F5053}$ (CYP

nomenclature: CYP1190B1) could be expressed in *E. coli* BL21 (DE3), while soluble Nas$_{S1868}$ (CYP nomenclature: CYP1190B1) could only be expressed in *Mycobacterium smegmatis* MC$^2$ 155.

Using an in vitro assay employing the electron-transport system (Fd and FdR) from spinach, Nas$_{S1868}$ can convert cW$_L$-P$_L$ into Asp-A (Fig. 2b, trace II; NMR and HRMS data see Supplementary Fig. 2, Supplementary Fig. 3 and Supplementary Table 3); a similar observation was also made by Li et al.[11]). Further, the in vitro assay confirmed that Nas$_{F5053}$ could produce NAS-C (47.4%), Asp-A (44.4%), as well as a minor product NAS-B (8.2%) (Fig. 2b, trace I). This catalysis profile is well correlated with the sequence alignments of Nas$_{F5053}$, NascB, Nas$_{S1868}$ and NasbB. Based on the sequence alignments, Nas$_{F5053}$ can be viewed as a chimeric form of NascB and Nas$_{S1868}$. The first 112 residues of Nas$_{F5053}$ are exactly the same as NascB (Fig. 2a). Except for mismatches in residues 125, 219, and 305, the C-terminal 273 residues of Nas$_{S1868}$ and Nas$_{F5053}$ are identical (Fig. 2a). This simple protein chimera strongly implies that the N-terminal portion of Nas$_{F5053}$ contains residues that are involved in the

generation of NAS-C, while the C-terminal part of $Nas_{F5053}$ harbors residues that are involved in the generation of ASP-A. Therefore, $Nas_{F5053}$, NascB, and $Nas_{S1868}$ provide a suitable portfolio of P450 to reveal the relationships between enzyme sequences and the corresponding products.

**Identifying the keys residues which determine the regio-configuration and stereo-configuration.** Next, to identify the roles of critical residues in regulating and controlling the product profiles of the P450 enzymes, a series of protein variants was generated with mutations in the N-terminal and C-terminal portions of $Nas_{F5053}$ and NascB, and the resultant mutant proteins were tested by enzyme assays. In the N-terminal part of $Nas_{F5053}$, we converted the following residues to their corresponding amino-acids in $Nas_{S1868}$: V44A-T49A-K52E (triple mutant), and P40H, Q65I, G84A, A86G, and I87V (point mutants). The in vitro enzyme assays of the $Nas_{F5053}$ mutants (P40H could not be tested as it was insoluble) using $cW_L-P_L$ showed that the simultaneous mutation of V44A, T49A, and K52E and single mutation of G84A or I87V imposed a slight change to the ratio of NAS-C and ASP-A (Supplementary Fig. 4, trace I-IV). Similarly, none of the eight point-mutations (P40H, V44A, T49A, K52E, S57P, F59L, G84A, and I87V) in the N-terminal portion of NascB were able to engineer NascB to produce ASP-A (Supplementary Fig. 4, trace V–XII). These eight positions were thus not investigated further. On the other hand, the single mutations Q65I and A86G in $Nas_{F5053}$ dramatically reduced the production of NAS-C (Fig. 2b, trace IV, V). Furthermore, the double mutation A86G-Q65I almost abolished the production of NAS-C, leaving ASP-A as the only detectable product (Fig. 2b, trace VI). These results clearly indicate that Q65 and A86 are the two crucial residues in $Nas_{F5053}$ that that direct the enzyme to produce NAS-C.

In the following steps, the same mutations (Q65I and A86G) were introduced into NascB. The Q65I mutation in NascB also impacted on the production of NAS-C; it decreased the production of NAS-C and increased the production of ASP-A from none to reach the NAS-C to ASP-A ratio of 5:7 (Fig. 2b, trace VIII). The single mutation A86G in NascB had a negligible effect on the production of ASP-A, whereas it neutralized the effect of Q65I in the A86G-Q65I double mutant, which is opposite to the synergistic effect observed in $Nas_{F5053}$ (Fig. 2b, trace IX, X). The contrasting effect of Q65 and A86 in NascB, as compared to $Nas_{F5053}$, prompts us to hypothesize that more residues in the C-terminal portion of NascB contribute to regulating the production of NAS-C and ASP-A. Because the C-terminal portion of NascB exhibits significant sequence differences, compared to $Nas_{S1868}$, the C-terminal part of NascB was divided into eight fragments (Supplementary Fig. 5). Each of the eight fragments was then replaced by the corresponding fragment in $Nas_{S1868}$, with Q65I and A86G mutations already in place. Each of eight NascB mutants was purified and enzyme assays revealed that the seventh fragment (fragment-7, carrying five mutations: H280Y, A284S, A287S, A288V, and L298I) almost abolished the production of NAS-C and generated ASP-A as the sole product (Fig. 2b, trace XI and Supplementary Fig. 6).

Point mutants were made to identify the effect of every single mutation in fragment-7 on the product profile. All five single point mutations produced more NAS-C than ASP-A (Supplementary Fig. 7, trace I–V), indicating more than one-substitution in fragment-7 is required to make ASP-A as the predominant product. We therefore restored, one by one, each of five-point mutations in the fragment-7 to its wild-type amino-acid of NascB, and observed that S284A or V288A counteracted the overall effect of five-point mutations in fragment-7 most

(Supplementary Fig. 7, trace VI-X). The results suggest that S284 and V288 are the two critical residues for the generation of ASP-A, while A284 and A288 are essential for the generation of NAS-C. Finally, the combination of Q65I-A86G-A284S-A288V in the NascB quadruple mutant was confirmed to make ASP-A as the major product (Fig. 2b, trace XII). In the case of $Nas_{F5053}$, the single mutations S284A or V288A significantly reduced the production of ASP-A, while the production of NAS-C was unaffected; and the double mutation S284A-V288A almost completely abolished the production of ASP-A (Fig. 2b, trace XIII–XV), hence unequivocally confirming the crucial role for these residues in determining the selective production of ASP-A and NAS-C. In addition to impact on the production of ASP-A and NAS-C, we also observed a reduced production of NAS-B in $Nas_{F5053}$-Q65I, $Nas_{F5053}$-A86G, and $Nas_{F5053}$-S284A (Fig. 2b, trace IV, V, XIII). Therefore, these residues in the four positions of 65, 86, 284, 288 apparently play the determining role in controlling regio-specificity and stereo-specificity of the generation of different frameworks in bacterial HDTKPs.

**Saturation mutagenesis of key residues to create different regio-specificities and stereo-specificities.** Considering that the reaction specificity of P450s can be regulated by only four key residues, we hypothesized that the creation of other frameworks with different regio-selectivities and stereo-selectivities is possible through engineering these four sites. Thus, the four key residues in $Nas_{F5053}$ (Q65, A86, S284, V288) were chosen simultaneously for NNK-based saturation mutagenesis. The mutated plasmids were transferred into GBdir-T7 E. coli containing spinach Fd and FdR, a whole-cell biocatalysis system we developed previously[6]. A small library of four hundred colonies was selected and assayed using $cW_L-P_L$ as a substrate. As expected, the production of NAS-B was significantly improved in some mutants. Among them, the mutant $Nas_{F5053}$-A86K-V288P not only yielded the highest ratio of NAS-B/ NAS-C (Fig. 2b, trace XVI, and Supplementary Fig. 8) but also produced another HTDKP product. Interestingly, $Nas_{F5053}$-Q65P-A86W-S284C also produced such compound, instead of NAS-B, in a ratio of 6:4 relative to NAS-C (Fig. 2b, trace XVIII). NMR and MS analyses identified this product, here we named as NAS-E; it contains a C3-aryl pyrroloindoline framework with a C3-C7' linkage and 2S-3R stereo-configuration (the type II HTDKP) (Fig. 1; NMR and HRMS data see Supplementary Fig. 9, Supplementary Fig. 2 and Supplementary Table 4). In order to validate this structure, we also synthesized it according to the reported total chemical synthesis strategies[14], and the comparison of the HPLC and NMR data of the synthetic compound with NAS-E unequivocally confirmed our proposed NAS-E structure (Fig. 2b, trace XVII and Supplementary Fig. 10). Cumulatively, both the production of NAS-E and the significant improvement in the yield of NAS-B further provide a compelling evidence that the four identified key residues control the regio-specificity and stereo-specificity of $Nas_{F5053}$ catalyzed-reactions.

**Crystal structures of NasF5053 and re-engineered mutants in complex with substrates.** To further understand the molecular basis of product diversity of $Nas_{F5053}$ and its homologs, we determined high-resolution structures of wild-type $Nas_{F5053}$ in its substrate-free (PDB ID 6W0S, Supplementary Fig. 11) and substrate-bound (PDB ID 6VXV) forms (Fig. 3a), by X-ray crystallography (Supplementary Table 5). $Nas_{F5053}$ adopts the prism-like fold characteristic for P450s, consisting of a large domain of 10-helices (C-L) and a small domain of four α-helices (A, B, B', and K') and three β-sheets (strands β1-1 to 4, β2-1 to 2, and β3-1 to 2) (Supplementary Fig. 11). The prosthetic heme group is bound at the crevice formed between helices I and L. Its

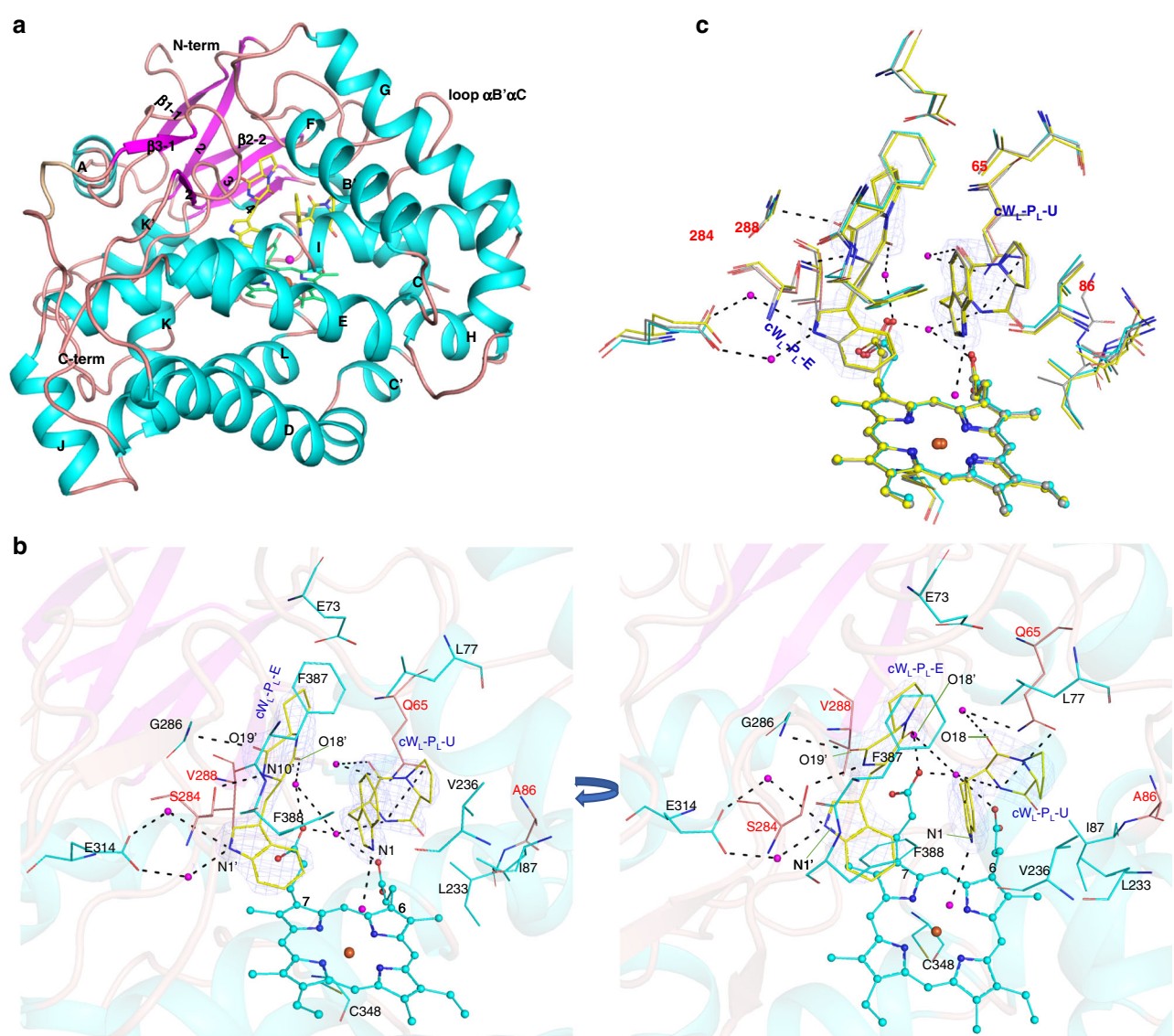

**Fig. 3 Crystal structures of Nas$_{F5053}$ and its mutants. a** Cartoon representation of the structure of Nas$_{F5053}$ bound to cW$_L$-P$_L$. Elements of secondary structure and the N/C-termini are labeled; α-helices are shown in cyan and β-strands in magenta. The iron in the heme is shown as a brown sphere and water molecules are displayed as magenta spheres. Other parts of the heme and cW$_L$-P$_L$ are displayed as green and yellow sticks, respectively. **b** A representation of the active site of Nas$_{F5053}$ in complex with cW$_L$-P$_L$-E and cW$_L$-P$_L$-U shown as yellow sticks. Oxygen and nitrogen atoms are shown in red and blue, respectively. The heme is displayed in cyan ball-and-stick representation, with the iron presented as a brown sphere. Nas$_{F5053}$ residues are colored in cyan. Left: "side" view of the active site; right, "top" view. Probable H-bonds between Nas$_{F5053}$, cW$_L$-P$_L$, heme propionate and water molecules (magenta spheres) are indicated as dotted lines. Four critical residues (Q65, A86, S284, and V288) are highlighted in red and shown as orange sticks. **c** Superposition of the active sites of Nas$_{F5053}$ (cyan sticks), Nas$_{F5053}$-Q65I-A86G (gray sticks), and Nas$_{F5053}$-S84A-V288A (yellow sticks) bound to cW$_L$-P$_L$-E and cW$_L$-P$_L$-U. The locations of four critical residues (65, 86, 284, and 288) are highlighted. The three complex structures are nearly identical, except for the mutated and adjacent residues. The cW$_L$-P$_L$-E and cW$_L$-P$_L$-U substrates in the NasF5053 complex structure are surrounded by Fo–Fc electron density omit map, which is calculated after 20 cycles of refinement in the absence of the ligands and contoured at 2.0 σ level (blue mesh).

heme iron is coordinated by the axial ligand Cys348 in helix L. At the distal side of the heme, the iron is coordinated by a water molecule (Supplementary Fig. 11), consistent with the EPR data for NascB[6] and CYP121[15] that water is coordinated predominantly to low-spin Fe (III).

Comparison between the substrate-free and substrate-bound Nas$_{F5053}$ structures reveals binding of substrates only invokes minimal conformational changes, with a root-mean-square deviation (RMSD) of 0.362 Å (for 388 Cα atoms) between the two forms (Supplementary Fig. 12). Instead, substrate binding is associated with rearrangements of some of the residues lining the substrate-binding cavity. Upon substrate binding, the side-chains of both D85 and E73 rotate along the Cα-Cβ axis away from the binding site, to accommodate the substrates. Q65 undergoes a 2.0 Å shift (measured on the Cδ atom) toward the binding site, to interact with one of the substrate molecules (cW$_L$-P$_L$-U; see below). Notably, Q65, D85, and E73 all reside in the long αB'-αC loop.

In the substrate-bound Nas$_{F5053}$ structure, two cW$_L$-P$_L$ molecules (cW$_L$-P$_L$-E and cW$_L$-P$_L$-U; E and U indicate extended and U-shaped, respectively) are present in the binding site, with full occupancy. cW$_L$-P$_L$-E adopts an extended conformation and forms multiple contacts with the heme group, loop β3-1-β3-2, loop αB'-αC, and loop αK-β1-4. F387 and L77 form hydrophobic

interactions with the proline portion of extended $cW_L$-$P_L$-E. Formation of hydrogen bonds is observed between S284 and the backbone amide nitrogen of G286, with N10' and O19' of the substrate, respectively. O18' and N1' are indirectly in contact with 7-propionate of the heme and E314, respectively, mediated via hydrogen bonding with water molecules. The hydrophobic side of V288 also protrudes towards 7-propionate of the heme and indole ring in $cW_L$-$P_L$-E (Fig. 3b).

On the other side, $cW_L$-$P_L$-U is mainly in contact with the heme, αI, αB' and long loop αB'-αC, including a T-shaped stacking interaction network between the F388 side-chain and the indole rings of both $cW_L$-$P_L$-E and $cW_L$-$P_L$-U. The DKP ring of $cW_L$-$P_L$-U is further restrained by the side-chain of Q65 and extensively stabilized by secondary interactions with water, 6-propionate of the heme, N10, O19, O18, the side-chain amide of Q65 and the backbone NH of A86. Multiple hydrophobic interactions are also observed with residues lining the binding site, including V236, L233, I87, and Q65. These interactions therefore force $cW_L$-$P_L$-U into a U-shaped folded conformation, bringing the indole and prolyl entities into close proximity (Fig. 3b). Notably, the folded conformation of $cW_L$-$P_L$-U brings its C2 and N10 into close contact (3.2 Å distance), making the intramolecular cyclization between $W_L$ and $P_L$ in $cW_L$-$P_L$-U possible. Importantly, the indole ring of $cW_L$-$P_L$-U is positioned perpendicular to the heme group plane, with N1 forming a hydrogen bond with the heme-ligating water molecule (Fig. 3b), consistent with the initial step of N-deprotonation reaction by P450 compound I[6]. The indole rings from the two substrate molecules also form a T-shaped stacking interaction with each other. Hence, the complex structure between $Nas_{F5053}$ and its substrate reveals a sophisticated orchestrated enzymatic environment where the heme, two identical substrates but different conformations, and the residues lining the substrate-binding cavity, are intimately interwoven.

However, our wild-type $Nas_{F5053}$ crystal structures in complex with $cW_L$-$P_L$ could not explain how $Nas_{F5053}$ produced two different products: NAS-C and ASP-A. To explain product selectivity, we therefore determined two more $cW_L$-$P_L$ substrate-bound crystal structures, of the mutants $Nas_{F5053}$-Q65I-A86G (PDB ID 6VZA) and $Nas_{F5053}$-S284A-V288A (PDB ID 6VZB). Comparisons among the three substrate-bound structures showed that all substrate-interacting residues, the heme, and the two substrates superimpose well (RMSD in this region between any two structures <0.26 Å; Fig. 3c), except for the mutated residues and the adjacent residues such as K289 and I87. These identical crystal structures indicates a common starting conformation for the reactions. To characterize NasF5053-catalyzed reactions further, we performed UV-Vis spectroscopic analysis and molecular dynamics (MD) simulations, to delineate the mechanism of regio-, stereo-selectivity and product profile regulation in NasF5053 and its re-engineered variants.

**Spectroscopic characterization of $cW_L$-$P_L$ binding to $Nas_{F5053}$.** We measure UV-Vis absorption and difference spectra to probe the interaction in solution between $cW_L$-$P_L$ and each of three enzyme variants, i.e., $Nas_{F5053}$, $Nas_{F5053}$-Q65I-A86G, and $Nas_{F5053}$-S284A-V288A. Binding of $cW_L$-$P_L$ to $Nas_{F5053}$ and its double mutants are all shifting a major Soret band from 418 nm to 387 nm, associating with the transition of the heme iron from the low spin (LS) to high spin (HS) state[16]. This transition, however, is not complete because a small but significant fraction of LS signal still remains even in the saturating $cW_L$-$P_L$ concentration (Supplementary Fig. 13).

Then the difference spectra are used to calculate the spectral variations with OriginPro software. The plotting of the spectral variation as a function of $cW_L$-$P_L$ concentration is fitting to a rectangular hyperbola curve, yielding a binding constant of 11.6 ± 2.1 μM for the interaction between $cW_L$-$P_L$ and wild-type $Nas_{F5053}$, 25.6 ± 1.0 μM for $cW_L$-$P_L$ with $Nas_{F5053}$-Q65I-A86G and 4.81 ± 0.26 μM for $cW_L$-$P_L$ with $Nas_{F5053}$-S284A-V288A (Supplementary Fig. 13). Data fitting to a rectangular hyperbolic shape also models the case of CYP121 with single substrate[15], suggesting that two $cW_L$-$P_L$ substrates with $Nas_{F5053}$ lack cooperativity for binding and catalysis. This assertion is further supported by a two-ligands complex structure where $cW_L$-$P_L$ occupies one site and $cW_L$-$P_L$ occupies the other site, which is reported in a published on-line research paper[17] when we are revising our manuscript.

**Molecular dynamics analysis**. To characterize $Nas_{F5053}$-catalyzed reactions further, we performed molecular dynamics (MD) simulations with Amber (Supplementary Fig. 14), to delineate the mechanism of regio-selectivity, stereo-selectivity, and product profile regulation in $Nas_{F5053}$ and its re-engineered variants. MD simulations were performed particularly to analyze the conformational changes associated with the proposed $cW_L$-$P_L$-U radical (Int1, Fig. 4) at the compound II stage[6]. The Q65-A86 and S284-V288 patches orchestrate the regio-specificities and stereo-specificities by distinct mechanisms. The Q65-A86 patch is involved in regulating the motion of the long loop αB'-αC, where Q65 and A86 reside at its two ends. The conformation of the αB'-αC loop influences the conformation of the $cW_L$-$P_L$-U radical. Based on the MD of native $Nas_{F5053}$, the $cW_L$-$P_L$-U radical rotates anticlockwise along the axis of N1-Fe (IV)-OH, until two indole rings of the two substrates are almost in a plane. The Q65I and A86G mutations result in a shift of the αB'-αC loop away from $cW_L$-$P_L$-U radical (Fig. 4a). The consequent relaxation of the restraints on Int1 unfolds Int1 (the distance between N10 and C2 is approximately 4.7 Å in most distance distributions; Fig. 4d). The results exclude the intramolecular cyclization of the $cW_L$-$P_L$-U radical to form a pyrroloindoline, without affecting the formation of ASP-A; this observation is consistent with our data that $Nas_{F5053}$-Q65I-A86G exclusively produce Asp-A.

The mutations of S284-V288 regulate regio-selectivity and stereo-selectivity by adjusting the relative positions of the two substrates and their conformations. Given that S284 and V288 contribute to lining the binding pocket for $cW_L$-$P_L$-E, mutations to less bulky Ala residues create space for $cW_L$-$P_L$-E to move towards the heme. This movement disturbs the interactions with the $cW_L$-$P_L$-U radical, and in turn pushes away the DKP and propyl rings of $cW_L$-$P_L$-U towards αI (orange sticks in Fig. 4b). This movement rigidifies the DKP ring of the $cW_L$-$P_L$-U radical as evidenced by decreased root-mean-square fluctuations (RMSFs) (Fig. 4f). According to MD, a positive C2-C3-C12-N10 dihedral angle between positions N10 above C2 (i.e., N10 attacks the Re face of the indole ring), generating an intermediate that leads to NAS-B, while a negative dihedral angles leaves N10 beneath C2 (i.e., N10 attacks the Si face of the indole ring), producing a different intermediate that leads to NAS-C. In the native form, there is a ~4–5 times higher probability for this dihedral angle to be negative (leading to NAS-C) than positive (leading to NAS-B), which is consistent with the experimental data that $Nas_{F5053}$ produces more NAS-C than NAS-B. The dihedral angle can only be negative in the S284-V288 mutant, echoing that this double mutant can catalyze the formation of only NAS-C (Fig. 4e). MD also shows that the probability of the C2-N10 distance in the wild-type protein being >4 Å or <4 Å is almost equal, which means that native $Nas_{F5053}$ could catalyze the formation of the products either requiring or escaping intramolecular cyclization. In the S284-V288 mutant, however, this

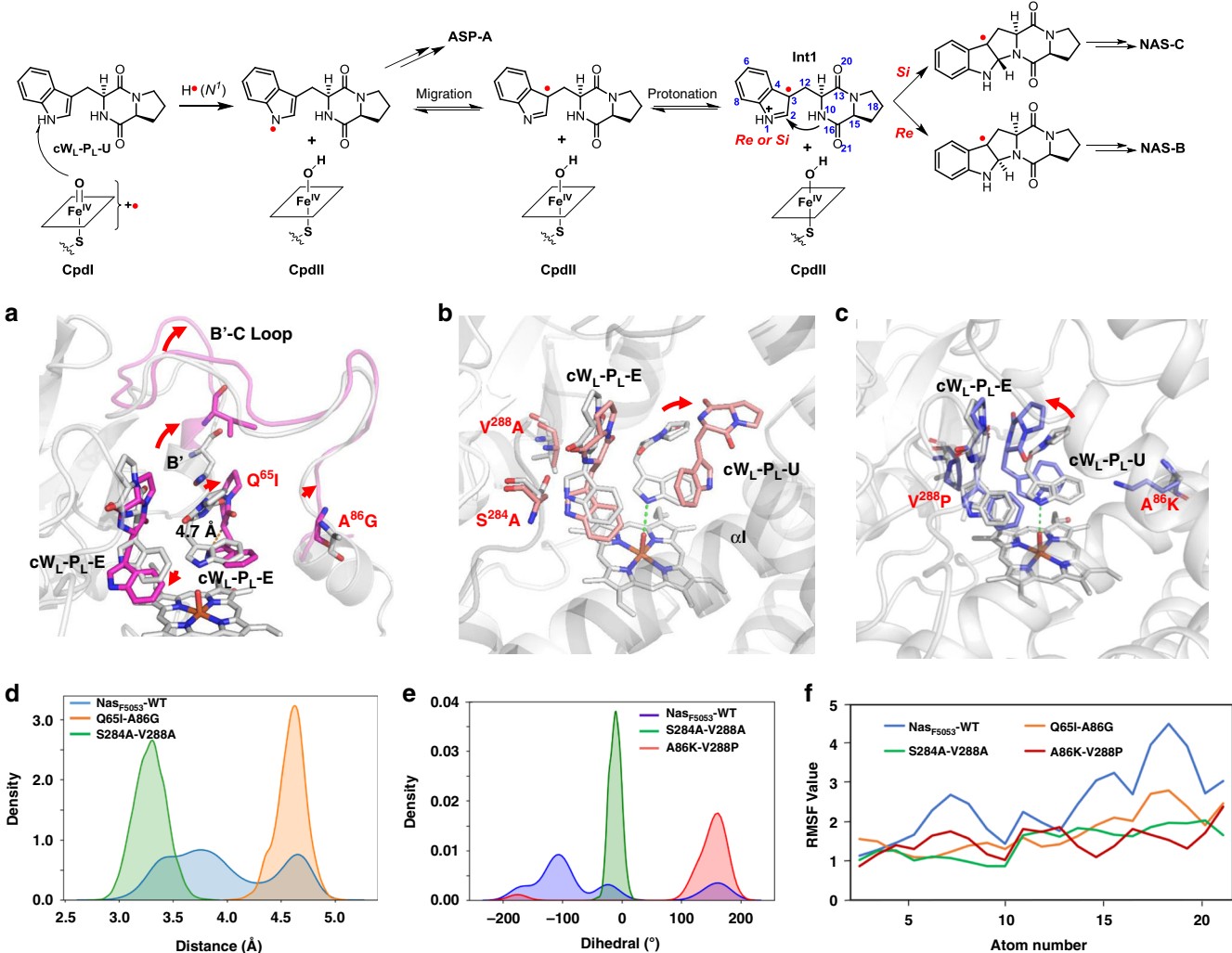

**Fig. 4 Molecular dynamics (MD) simulations of Nas$_{F5053}$ (WT), Nas$_{F5053}$-Q65I-A86G, Nas$_{F5053}$-S284A-V288A and Nas$_{F5053}$-A86K-V288P in the presence of the substrate cWL-P$_L$-E and the cW$_L$-P$_L$-U radical (Int1).** CpdI and CpdII are compound I and compound II, respectively. In a cartoon representation, selected active site residues are shown as sticks. **a** Superposition of WT Nas$_{F5053}$ (gray) and Nas$_{F5053}$-Q65I-A86G (pink). **b** Superposition of WT Nas$_{F5053}$ (gray) and Nas$_{F5053}$-S284A-V288A (salmon pink). **c** Superposition of WT Nas$_{F5053}$ (gray) and Nas$_{F5053}$-A86K-V288P (cyan). **d** Distances between N$^{10}$ and C$^2$ of the cW$_L$-P$_L$-U radical in WT Nas$_{F5053}$ (blue), Nas$_{F5053}$-Q65I-A86G (orange), and Nas$_{F5053}$-S284A-V288A (green). **e** C$^2$-C$^3$-C$^{12}$-N$^{10}$ dihedral angles of the cW$_L$-P$_L$-U radical in WT Nas$_{F5053}$ (blue), Nas$_{F5053}$-S284A-V288A (green), and Nas$_{F5053}$-A86K-V288P (red). **f** RMSF values of the cW$_L$-P$_L$-U radical in WT Nas$_{F5053}$ (blue), Nas$_{F5053}$-Q65I-A86G (brown), Nas$_{F5053}$-S284A-V288A (green), and Nas$_{F5053}$-A86K-V288P (red). For atom numbers, see the cW$_L$-P$_L$-U radical (int1) in the top panel.

distance is fixed between 3.0 and 3.5 Å, making intramolecular cyclization inevitable (Fig. 4d).

The re-positioning and conformational changes of the substrates can also be achieved by the combined mutations in both the Q65-A86 and S284-V288 sites, such as the A86K-V288P double mutant. Opposite to its wild-type form, the A86K-V288P mutant produces NAS-B as the major product and NAS-C as the minor product. The long side-chain of K86 protrudes towards cW$_L$-P$_L$-U radical and drives its rotation and shift towards cW$_L$-P$_L$-E. On the other side, the V288P mutation compresses the active site, slightly pushing and rotating cW$_L$-P$_L$-E (cyan sticks in Fig. 4c). The dual changes of cW$_L$-P$_L$-U radical and cW$_L$-P$_L$-E reach a conformation where the DKP ring of cW$_L$-P$_L$-U radical becomes more rigid. In such a conformation, the C2-C3-C12-N10 dihedral angle is positive with high probability, favoring the attack of N10 to the Re face of the indole ring, to generate an intermediate leading to NAS-B. This is accompanied by a low probability event, where the dihedral angle is negative to allow for the formation of an intermediate leading to NAS-C. The sign

distribution of the dihedral angle is supported by the product profile of the A86K-V288P double mutant.

## Discussion

Cytochrome P450 (CYP) enzymes are among the most exquisite and versatile biocatalysts in nature to synthesize and modify natural products[18,19]. P450s and their engineered variants are continuously exploited as biocatalysts to functionalize natural products or potential drug leads[20]. P450-catalyzed reactions can be broadly categorized into two groups: common and unusual[21]. Common P450 reactions generate minor structural alterations, such as C–H, N–H hydroxylation, and epoxidation on C=C double bonds. The mechanisms for those reactions are clear and represented by a canonical P450 catalytic cycle while the mechanisms for unusual P450 reactions are often unknown or elusive. Along with uncharacterized mechanism for uncharted chemistry, unusual P450 reactions may catalyze an enigmatic and/or dramatic structural transformation. Those features of unusual P450 reaction are of special research interests.

As an unusual P450-catalyzed reaction, the reaction of NascB was assumed to involve radical generation at N1 and migration, intramolecular Mannich reaction to form the pyrroloindoline C3 radical, and radical addition to the other molecule of DKP to form the HTDKP framework[6]. Although our previous DFT calculations and experiments preferred the N1-initiation over N10-initiation mechanism[6], there was a lack of direct proof. Based on the crystal structures, we can now clearly see that the N1 of $cW_L$-$P_L$-U is indeed much closer than the N10 to the heme-ligating water molecule (Fig. 3b). In addition, the $cW_L$-$P_L$-U is in a U-shaped, folded conformation. Its indole and prolyl entities are close to each other, providing a viable distance for the intramolecular Mannich reaction to form the pyrroloindoline C3 radical. As $Nas_{F5053}$ shows no structural evidence to accommodate the second copy of the pyrroloindoline C3 radical, the radical dimerization mechanism proposed in fungal TDKP biosynthesis can also be excluded[7,8]. Furthermore, the three well superimposed complex structures (Fig. 3c) suggest a conserved starting conformation and reaction initiation steps in the formation of ASP-A, NAS-C, and NAS-B, although differentiating conformational dynamics of substrates develop in wild-type $Nas_{F5053}$ and the three mutants, leading to the formation of different products. Therefore, all our structural evidence solidly supports the assumed reaction mode of HTDKPs[6]. Except for the type V HTDKP formation through a N1-radical addition, the intramolecular and intermolecular radical cascade mechanism[6] thus can be rationalized to be a common paradigm for the biosynthesis of other bacterial HTDKPs.

Our dynamics simulation analyses indicate that the stereo-specificity and regio-specificity of P450 is indeed controlled by a sophisticated interaction of the substrates with the protein. This observation is consistent with the previous results for NascB, which can generate various HTDKP products with type I–IV frameworks upon feeding different substrates[6]. In contrast to the substrate-based approach, engineering the specificity conferring-residues is more appealing for biocatalysis to generate structural diversity of HTDKPs. Although the outcome of the reaction specificity cannot be readily predicted solely based on the crystal structures, the identified four specificity-conferring residues can serve as targets for protein engineering. Through screening a small library of mutations on the four residues, the product specificity of $Nas_{F5053}$ was able to be shifted between different frameworks, which enables $Nas_{F5053}$ to predominantly produce NAS-B, NAS-C, ASPA, or even NAS-E. This approach makes it very convenient for biocatalysis to efficiently produce the desired types of HTDKPs. In addition to the five identified types of HTDKPs, engineering the specificity-conferring residues also has the potential to generate diverse frameworks; screening more mutants for finding different specificities is currently in progress.

Besides the regio-specificity and stereo-specificity to generate frameworks, the limited tolerance of P450s for substrates is another factor that restricts their application. Previously, we found NascB has a very limited freedom in accepting substrates at the $cW_L$-$P_L$-E site[6]. From the structure, $cW_L$-$P_L$-E is surrounded by the bulky residues E73, F387, and L77; especially E73 is very close to the substrate. These residues, constituting the "ceiling" of the pocket, may form a constraint that hinders accepting bulkier substrates. In the P450 dimerases for a few of HTDKP-like products, which contain heterodimerized nucleobase-DKP frameworks[22–24], E73 is replaced by the larger residue Tyr in GutD and $P450_{NB5737}$[22,23]. As nucleobases are smaller than DKPs, this bulky residue may act as a gatekeeper, to restrict the second copy of DKP entering the pocket and to force the enzyme to catalyze a hetero-dimerization between the nucleobase and DKP. Therefore, engineering these residues may be able to control the space of the binding pocket and subsequently enable the enzymes to accept either larger or smaller substrates in the prolinyl position of $cW_L$-$P_L$-E, and currently such attempts are in progress. At the bottom of the binding pocket, the U-shaped molecule has more freedom as observed in our previous study[6], by extending its proline moiety to the tunnel entrance lined by another two gatekeeper residues, V236 and L77. Engineering these two residues has the potential for further broadening of the substrate scope in the "bottom" cavity. By combining the engineering in reaction specificity and pocket space, the P450 reactions are believed to be able to generate more varied molecular diversity of HTDKPs.

The reaction specificities of P450s are determined by the sophisticated and orchestrated enzymatic environments and therefore it is difficult to identify the specificity-conferring residues solely from the crystal structures, especially for Ala86, which is ~6 Å away from $cW_L$-$P_L$-U (Fig. 3b). Through repeated construction and evaluation of sequence chimeras, we provided a strategy to decipher the sequence-product relationships of HTDKP-producing P450 enzymes. This approach proves to be effective in identifying pivotal residues governing product specificity between two or more homologous proteins. Based on these discoveries, we were able to alter the P450s' specificity through protein engineering. For enzymatic reactions with complicated catalytic mechanisms, relying solely on structural analysis can easily miss important information. Therefore, it is better to incorporate the investigation of the sequence-product relationships and our approach provides an option for this purpose.

In conclusion, through discovery, identification, and functional characterization, we have identified a suite of P450s (NasbB, $Nas_{F5053}$, and $Nas_{S1868}$) that share high sequence similarities but generate unique overlapping product profiles across all the five types of bacterial dimeric DKP frameworks. Our systematic mutagenesis studies on the promiscuous $Nas_{F5053}$ and the versatile NascB identified four key resides, Q65, A86, S284, and V288, which play critical roles in controlling product regio-configurations and stereo-configurations. We demonstrate that the engineering of these residues is able to alter the product ratio and even generate an interesting framework, which has not previously been observed for the substrate $cW_L$-$P_L$. To obtain insights into the structural basis for regio-specificity, stereo-specificity, and chemical versatility, we further determined high-resolution crystal structures of wild-type $Nas_{F5053}$ in its substrate-free and substrate-bound form, and of two $Nas_{F5053}$ mutants (Q65I-A86G and S284A-V288A) in their substrate-bound forms. The binding mode of $cW_L$-$P_L$ revealed by the complex structures supports the previous proposed intramolecular and inter-molecular radical cascade addition mechanism. Molecular dynamics simulations were employed to uncover the specificity-conferring mechanism of these residues, based on the crystal structures. Therefore, our biochemical, structural, and computational characterizations across this representative group of HTDKP-forming P450s provide a clear mechanism of how these sophisticated catalytic mechanisms take place, which expands our knowledge on the chemical diversity of cytochrome P450s-catalyzed natural products and enables the rational engineering of this group of P450s and other homologs to obtain different HTDKP frameworks.

While this manuscript was undergoing revision after review, Shende and Co-workers published the structural and functional characterization of NzeB[17], the synonym of $Nas_{F5053}$. Their structural data are consistent with our data. The active site residues that they identified are also covered by four key residues revealed in our manuscript.

## Methods

**Protein expression, purification, and enzyme assay.** P450 genes with codon optimized for *E. coli* were cloned into pET28a (*nas_{F5053}*) and pMS1 (*nasbB* and

$nas_{S1868}$), which were overexpressed in *E. coli* BL21 (DE3) and *M. smegmatis* mc$^2$ 155, respectively (Supplementary Fig. 15). The in vitro biochemical reactions using all the P450s mentioned in this study were performed in a 100 μL reaction system containing 0.1 μM P450, 1 mM cW$_L$-P$_L$, 1 μM spinach ferredoxin (Fd), 1 μM ferredoxin reductase (FdR), 2 mM NADP$^+$, 2 mM glucose, and 2 mM glucose dehydrogenase (GDH) in 50 mM HEPES buffer, 100 mM NaCl, at pH 7.5. After incubating at 4 °C for 24 h, the reactions were quenched and extracted with ethyl acetate (2 × 200 μL). Then the combined organics were concentrated in vacuo, which were re-dissolved in HPLC-graded methanol and the resulting solutions were filtered through 0.45 μM membrane and finally analyed by UHPLC-MS. A Diamonsil (C18, 2 μm, 2.1 × 50 mm, Shim-pack GIST) was used with a flow rate at 0.3 mL min$^{-1}$ and a PDA detector over a 23 min gradient program with water (eluent A) and methanol (eluent B): $T = 0$ min, 40% B; $T = 10$ min, 40% B; $T = 15$ min, 70% B; $T = 18$ min, 40% B; $T = 23$ min, 40% B.

**Protein crystallization and crystal structure determination**. Initial crystals were obtained in 0.2 M CaCl$_2$, 20% (w/v) polyethylene glycol (PEG) 3350, pH 7.5 at 20 °C using the hanging drop vapor diffusion technique with the addition of 5% glycerol to the protein stock. The initial crystals were subsequently crushed for seeding by using the Seed Bead Kit (Hampton Research). Final crystals were obtained using the micro-seeding technique in 0.2 M CaCl$_2$, and 22% (w/v) PEG 3350, pH 7.5 at 4 °C. Substrate-bound protein crystals were obtained by soaking the substrate-free crystals in the mother liquor containing 2.5 mM cW$_L$-P$_L$ (diluting from 50 mM stock solution in DMSO) for 24 h or co-crystallization after mixing 0.13 mM protein with 2.5 mM cW$_L$-P$_L$. Both methods produce identical complex structures. The complex structure from soaking was chosen for structural analysis and presentation, due to a better overall quality, including resolution, of the diffraction data collected from the soaked crystals.

Crystals were mounted onto CryoLoops (Hampton Research) and soaked in a cryoprotection solution containing 0.2 M CaCl$_2$, 22% (w/v) PEG 3350, pH 7.5, and 20% (v/v) glycerol prior to flash freezing in liquid. For the substrate-bound protein crystals, the cryoprotection solution also contained 2.5 mM cW$_L$-P$_L$. The X-ray diffraction data were collected at the Australian Synchrotron MX beamlines. The collected data were indexed and integrated using *XDS*[25] and scaled and merged using *Aimless*[26]. A partial initial model of the holo-structure was obtained by the molecular replacement technique with *Phaser* in *Phenix*[27] using the crystal structure of CYP121 from *Mycobacterium tuberculosis* (PDB accession code: 5WP2) as the search model. The initial model was improved by using the *Morph Model* tool in *Phenix*[28] and manually modified in *COOT*[29]. The substrate-bound structure was solved by the molecular replacement technique using the holo-structure as the search model. The structures were refined using *Phenix.Refine*[30] and manually modified in *COOT* iteratively. The graphic presentations of protein structures were prepared with Pymol.

**NMR spectroscopy**. The NMR spectra were recorded on a Bruker Avance III spectrometer at a $^1$H frequency of 400 MHz. Lyophilized samples (varying from 1 to 7 mg) were dissolved in 280 μL DMSO-*d6* (Cambridge Isotope) and all spectra were recorded at 25 °C (298 K). $^1$H and $^{13}$C resonances were assigned through the analysis of 1D-$^1$H, 1D $^{13}$C, 2D $^1$H–$^1$H ROESY, 2D $^1$H–$^{13}$C HSQC, and 2D $^1$H–$^{13}$C HMBC (optimized for long-range heteronuclear couplings of 6 Hz). $^1$H and $^{13}$C chemical shifts were calibrated with reference to the DMSO solvent signal (2.50 and 39.5 ppm for $^1$H and $^{13}$C, respectively). NMR experiments were processed with Bruker Topspin program (version 3.57) and analyzed with MestReNova software.

**Reporting summary**. Further information on research design is available in the Nature Research Reporting Summary linked to this article.

## Data availability

The sequences of the *nasbB*, *nas$_{S1868}$*, and *nas$_{F5053}$* reported in this work are available under existent accession numbers MW196742 (this hyperlink is currently on hold and will be released upon publication), WP_030881046.1 and WP_030888003.1 in Genebank, respectively. Their synthetic DNA sequences are listed in the supplementary information. All other data are also available upon request from the corresponding authors. The pdb coordination files for substrate-free F5053 and F5053 in complex with cW$_L$-P$_L$, Nas$_{F5053}$-Q65I-A86G in complex with cW$_L$-P$_L$ and Nas$_{F5053}$-S284A-V288A in complex with cW$_L$-P$_L$ were deposited in Protein Data Bank with the accession number of 6W0S, 6VXV, 6VZA, and 6VZB, respectively. Source data are provided with this paper.

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

## Acknowledgements

We thank Prof. Rob Capon and Dr. Zeinab Khalil for kindly providing the *Streptomyces* sp. CMB-MQ030 strain and the NAS-B standard; Prof. Xiaoyong Fan for the gift of the plasmid pMV406; Prof. Jiaoyu Deng for the gift of pMV206 and *M. smegmatis* mc$^2$ 155; Prof. Xiaoyan Cui for providing help in UV-Vis Spectroscopy analysis and Prof. Jeffrey

R. Harmer for critical discussion in UV-Vis titration. This work was supported in part by the NSFC (31770063 to X.Q.), National Key R&D Program of China (2018YFC1706200 to X.Q.), Shanghai Post-doctoral Excellence Program (2019193 to C.S.), The University of Queensland (UQ Early Career Researcher Grant UQECR1946973 to X.J.), Australian Research Council (ARC Laureate Fellowship FL180100109 to B.K.). We acknowledge the facilities, and the scientific and technical assistance of the Australian Microscopy and Microanalysis Research Facility at the Centre for Microscopy and Microanalysis, The University of Queensland and the macromolecular crystallography (MX) beamlines at the Australian Synchrotron, Victoria, Australia.

## Author contributions

X.Q., X.J., B.K., and Z.D. conceived this project; C.S., W.Z., W.T., H.P., and Zhi Lin performed the biochemical experiment; Z.L. and X.J. solved the crystal structures; X.Q., X.J., C.S., and Z.L. analyzed data; and X.Q., X.J., C.S., and Z.L. wrote the manuscript.

## Competing interests

The authors declare no competing interests.
