## [Peer Review File · Nature Communications]

REVIEWER COMMENTS

Reviewer #1 (Remarks to the Author):

The manuscript (Manuscript ID: NCOMMS-20-24724) by Chenghai Sun, et al. entitled "Molecular Basis of Regio- and Stereo-Specificity in Biosynthesis of Bacterial Heterodimeric Diketopiperazines" reported P450s (NasB, NasF5053 and NasS1868) that share high sequence similarities but generate unique overlapping product profiles across all the five types of bacterial dimeric DKP frameworks. Mutagenesis studies on the promiscuous NasF5053 and the versatile NascB identified four key residues, Q65, A86, S284 and V288, which play critical roles in controlling product regio- and stereo-configurations as well as engineering of these residues to alter the product ratio and even generate a novel framework, which has not previously been observed for the substrate cWL-PL. Analysis of crystal structures of wild-type NasF5053 in its substrate-free and substrate-bound form enables us to understand regio-, stereo-specificity and chemical versatility. The authors further determined high-resolution crystal structures of two NasF5053 mutants (Q65I-A86G and S284A-V288A) in their substrate-bound forms. The complex structures reveal a novel cWL-PL binding mode, supporting the previous proposed intramolecular and intermolecular radical cascade addition mechanism. The work is scientifically sound and will allow others to understand and exploit the enzymes and related pathways. This referee has reviewed very much similar manuscript submitted to JACS by different authors. The current manuscript should be better in terms of molecular dynamic analysis. Thus, the manuscript will be of great interest to the readers of Nature communications.

As such, the paper is recommended for publication in Nature communications as an Article after revision.

Questions:

1. Page12, line 431, This reviewer found unnatural reaction conditions, which the enzyme activity was conducted at 4°C. Usually, we often test around 20~30°C. In Supporting S39, line 378, it was still at 4°C for the reactions. While bacteria produced enzymes in this study, the enzymatic reactions should be incubated at 18°C (supporting S41, page 456) under this study. Why is the enzymatic reaction performed at 4°C?
2. Page 21, Figure 4 legend, Chemical structures of "Cpd I and CpdII" should be illustrated in this manuscript.
3. Page6, line 194, "NasF5053-A89K-V288P" should be changed to "NasF5053-A86K-V288P"
4. This reviewer is not able to understand differences between "Fdx" and "Fd" as well as "FdxR" and "Fdr" Should be clear.
5. Page19, line 634, "XII) NascB-Q65I-A86G-S284A-V288A" should be changed to "XII) NascB-Q65I-A86G-A284S-A288V"
6. Page19, line 635, "XVI) NascF5053-A89K-V288P" should be changed to "XVI) NascF5053-A86K-V288P"
7. Page21, Figure 4D and 4E, The authors should describe what the X- and Y-axis show.

Reviewer #2 (Remarks to the Author):

The manuscript reports the identification of new bacterial cytochromes P450 (namely NasF5053 and NasS1868) involved in the biosynthesis of heterodimeric diketopiperazines that are bioactive natural compounds with interesting pharmacological properties. Thanks to their characterization in terms of product formation and sequence alignments with other Nas proteins, different approaches of protein engineering were applied to the N- and C-termini of NasF5053 to identify the key residues involved in the preferential generation of NasC versus AspA. Moreover, its crystal structure in the substrate-free and -bound forms as well as the structure of two double-mutants

were solved at high resolution providing the starting point for MD simulations that allow to understand the molecular basis of the regio- and stereo-specificity of this enzyme. The work is highly interesting for a broad readership because it provides a beautiful example of how protein engineering combined to x-ray crystallography can be applied to understand the molecular basis of enzyme specificity. Different protein engineering approaches were applied at the N- and C-terminal of the protein in order to identify the key residues enabling specificity and the structural data strongly supported the functional ones.

In general, the manuscript is well written, but sometimes hard to follow especially for non-specialists in the field of cytochromes P450 and also because many different proteins and compounds are present. It can be suitable for publication on Nat. Comm. after a revision according to the following points to be addressed and suggestions.

Major points.

1. The crystal structure shows two molecules of the substrate in two different conformations in the active site of the enzyme. From the structural analysis, it looks like the co-presence of these two molecules, together with the protein-substrate interactions, is essential for specificity. It can be expected therefore that there is a cooperative effect for substrate binding and/or catalysis meaning that the second substrate molecule pushes the first one to adopt a different conformation. Have the Authors tested cooperativity for substrate binding and/or catalysis?
2. The water molecule coordinating the heme iron is present in the crystal structure in both the substrate-free and -bound forms. Is the substrate moving away the water molecule and inducing the low-to high spin transition? What do the spectra look like? If there is a transition, than it becomes easy to test cooperativity for substrate binding.
3. Figure 3B and 3C are not clear. The substrate molecules are not distinguishable from the protein. The style of Supplementary Fig. 3B should be adopted.

Minor points.

1. A paragraph introducing for non-specialists in the field the role of cytochromes P450 in the biosynthesis of natural bioactive compounds of pharmaceutical interest should be added with references to a couple of review (see for example doi 10.1039/C7NP00028F and doi 10.1016/j.tibs.2020.03.004).
2. Lines 54-65. A table summarizing the different proteins considered and their specificities can help the reader to follow. Another possible option is to add the name of the proteins producing the different compounds in Figure 1 with the percentage reflecting their specificities.
3. Line 144. Here the sentence is not clear. It should be clearly explained that the same mutations were introduced also in NascB.
4. Figure 4D-4E. Axis labels are missing.
5. The CC1/2 should be added to Supplementary Table 4.

Reviewer #3 (Remarks to the Author):

This is a nice manuscript entitled 'Molecular Basis of Regio- and Stereo-Specificity in Biosynthesis of Bacterial Heterodimeric Diketopiperazines' from the groups of Qu, Jia and Kobe. This paper encompasses a large and elegant body of work describing the structural characterization of three related cytochrome P450 enzymes from *Streptomyces* spp. that are involved in the production of heterodimeric tryptophan-containing diketopiperazines (HTDKPs). HDKTS are complex molecules that are very difficult to synthesise chemically and typically display a wide range of bioactive reactivity's. The P450 enzymes catalyse differing regio- and stereo-specific carbon-carbon bond coupling reactions to produce overlapping but diverse HDKTPs. The authors have determined that four key amino acids residues play a major role in conferring these regio- and stereo-specificities of the enzymes and subsequent products through a large mutagenic study creating both single and combinations of amino acid mutants as well as the generation of P450 chimeras. Saturation mutagenesis was then employed to explore these key positions and expand the product repertoire and specificities of these P450 enzymes and identified a novel naseseazine derivative NAS-E. High

resolution crystal structures were solved for the NasF5053 enzyme in the ligand free and substrate bound form as well as for two variants that each contained two of the four key selectivity amino acid substitutions. This study also utilised molecular dynamic simulations to model conformational changes that were not observed in the crystal structures and give an insight into reaction mechanisms of these enzymes.

This paper leads on from the author's previous work in this field (Tian et al. 2018) and displays a comprehensive study determining the specificities and mechanistic properties of this group of HTDKP forming enzymes as well as some interesting future perspectives. This work is of interest to the Nature Communications reader and brings together a large amount of work. I do question whether this work is novel enough to appeal to the broader interests' out with this specific field. However, this paper brings together a number of techniques with very well and shows very nicely how they can merge to answer mechanistic questions and facilitate enzyme engineering. I also like that the enzymatic products are clearly characterised both by biotransformation reactions and good MS and NMR data. Overall I think this is a very convincing study.

The manuscript is clearly written but could be perhaps a bit more concise. It would be useful to the reader to formally identify and name the different cytochrome P450 using CYP nomenclature (see <https://uthsc.edu/medicine/molecular-sciences/faculty-directory/nelson.php>) as the current nomenclatures can be a little confusing.

Could the authors clarify the difference between the strains of *Streptomyces* S1868 used in this study and *Streptomyces* NRRL S-1868 used in the study of Li and co-workers and their respective proteins NasS1868 and NasB, are they actually distinct?

The presence of two substrate molecules in the active site in differing conformations is not unprecedented. However, it would have been nice to see some solution data alongside this study to help confirm whether this is a functional or dynamic aspect or merely a result of soaking excess substrate into a pre-formed crystal and as such an artefact of the crystallography. The use of UV-vis binding titrations and/or ITC data would have shown if there was any cooperativity during substrate binding as well as give binding affinities for each enzyme variant.

Minor edits:

- Line 124: add "in the" so reads "mutations in the N-terminal and C-terminal portions of NasF5053 and NasB"
- Line 303: remove full stop following C2-N10
- Line 420: reword or remove "future" overexpressed" this terminology is confusing
- Line 423: add concentration of FdR
- Figure 4: Labels on axes would be useful
- SI line 228: remove "of the" so reads "on Luria-Bertani (LB) media agar plates or in Lemoco liquid media"

Best wishes, Dr Kirsty McLean

We thank the reviewers for their helpful inputs and comments in order to improve our manuscript. We do agree with all comments from reviewers and have addressed all the reviewers' comments, which are detailed in our responses below:

Reviewer #1

Comment 1: Page12, line 431, This reviewer found unnatural reaction conditions, which the enzyme activity was conducted at 4°C. Usually, we often test around 20~30°C. In Supporting S39, line 378, it was still at 4°C for the reactions. While bacteria produced enzymes in this study, the enzymatic reactions should be incubated at 18°C (supporting S41, page 456) under this study. Why is the enzymatic reaction performed at 4°C?

Response: For the enzymatic reaction, we have tested many temperatures including 4°C, 10°C, 16°C, 22°C, 30°C, 37 °C, 42°C and 50°C. Our results show that the P450s indeed exhibit the best activity at 4°C (still active at 50°C but very weak). Although enzymes are usually expressed and more active at higher temperature, in vitro enzymatic reactions are also influenced by enzyme stability. We conclude that 4°C is the best temperature among the assayed temperatures for the activity and enzyme stability of all tested P450s in our manuscript.

Comment 2: Page 21, Figure 4 legend, Chemical structures of “Cpd I and CpdII” should be illustrated in this manuscript.

Response: We have added the structures of Cpd I and Cpd II in Figure 4.

Comment 3: Page6, line 194, “NasF5053-A89K-V288P” should be changed to “NasF5053-A86K-V288P”

Response: This has been corrected. We have fixed all similar errors in the Manuscript, including the legends of Figure 2 and Figure 4.

Comment 4: This reviewer is not able to understand differences between “Fdx” and “Fd” as well as “FdxR” and “FdR” Should be clear.

Response: We are sorry for these typos. There is no difference between “Fdx” and “Fd” as well as “FdxR” and “FdR”. We have changed all “Fdx” to “Fd” as well as “FdxR” to “FdR”.

Comment 5: Page19, line 634, “XII) NascB-Q65I-A86G-S284A-V288A” should be changed to “XII) NascB-Q65I-A86G-A284S-A288V”

Response: We have corrected these errors.

Comment 6: Page19, line 635, “XVI) NascF5053-A89K-V288P” should be changed to “XVI) NascF5053-A86K-V288P”

Response: We have corrected this error.

Comment 7: Page21, Figure 4D and 4E, the authors should describe what the X- and Y-axis show.

Response: We have added the axis labels in Figure 4D and 4E.

Reviewer #2:

Major points.

Comment 1: The crystal structure shows two molecules of the substrate in two different conformations in the active site of the enzyme. From the structural analysis, it looks like the co-presence of these two molecules, together with the protein-substrate interactions, is essential for specificity. It can be expected therefore that there is a cooperative effect for substrate binding and/or catalysis meaning that the second substrate molecule pushes the first one to adopt a different conformation. Have the Authors tested cooperativity for substrate binding and/or catalysis?

Response: We highly appreciate this suggestion. This comment is related to

the question raised by Reviewer#3. As suggested by Reviewer#3, we used the UV-vis binding titrations to analyze any cooperative effect of substrates. The UV-Vis absorption differences of Nas_{F5053}, Nas_{F5053}-Q65I-A86G, or Nas_{F5053}-S284A-V288A against the substrate titration are all fitted to a rectangular hyperbolic shape (Supplementary Fig. 13) which models the example of CYP121 with a single substrate (ref15, Doi: 10.1073/pnas.0812191106), indicating that we could not observe any detectable cooperativity for substrate cW_L-P_L binding to Nas_{F5053} or its double mutants.

We have included those descriptions in a new section (Line 276-293) of the Results.

Comment 2: The water molecule coordinating the heme iron is present in the crystal structure in both the substrate-free and –bound forms. Is the substrate moving away the water molecule and inducing the low-to high spin transition? What do the spectra look like? If there is a transition, than it becomes easy to test cooperativity for substrate binding.

Response: Upon the addition of increasing concentration of cW_L-P_L, the major Soret band in Nas_{F5053} UV-Vis absorption spectra was shifted from 418 nm to 387 nm (Supplementary Figure 13). The spectral changes are attributed to the transition of the heme iron from the low spin to high spin state, accompanied by the displacement of the water molecule by the substrate. However, even at saturating substrate concentrations, there is still a small but significant fraction of LS signal persisting.

In our previous paper about NascB (Ref6, Doi: 10.1038/s41467-018-06528-z), we used EPR to measure the ferric heme signal. Upon the addition of two times more substrate, most of the ferric heme ion remains low spin, which indicates that the substrate is not moving away the water molecular that

coordinates the resting state heme iron. Considering that NascB shares 87% sequence identity with NasF5053, we assume that EPR spectra of Nas_{F5053} would be very similar to that of NascB.

Combining UV and the previous EPR spectra analysis, we conclude that the substrate will move away the water and induce the low to high spin transition of the heme iron. The population of low spin and high spin of the heme iron will depend on enzyme concentrations, substrate concentrations and binding constants. In the crystallization condition, apparently most of the heme iron stays in the low spin state and water coordinates with the heme iron.

Comment 3: Figure 3B and 3C are not clear. The substrate molecules are not distinguishable from the protein. The style of Supplementary Fig. 3B should be adopted.

Response: Supplementary Fig. 3B is not related to the illustration of crystal structure. We presume what the reviewer meant by “The style of Supplementary Fig. 3B” is the density map shown in Supplementary Fig. 11 B (original Supplementary Figure 10B). We have therefore included a light-colored density omit map surrounding the substrates to distinguish between the substrates and the protein in Figure 3B and 3C.

Minor points.

Comment 4: A paragraph introducing for non-specialists in the field the role of cytochromes P450 in the biosynthesis of natural bioactive compounds of pharmaceutical interest should be added with references to a couple of review (see for example doi 10.1039/C7NP00028F and doi 10.1016/j.tibs.2020.03.004).

Response: We have included a short paragraph (line 350-361) as in the section of the discussion describing the general features of P450 for natural product biosynthesis. The relevant references have been also cited.

Comment 5: Lines 54-65. A table summarizing the different proteins considered and their specificities can help the reader to follow. Another possible option is to add the name of the proteins producing the different compounds in Figure 1 with the percentage reflecting their specificities.

Response: We have added a new Supplementary Table 1 that summarizes the different specificities of each protein.

Comment 6: Line 144. Here the sentence is not clear. It should be clearly explained that the same mutations were introduced also in NascB.

Response: We added a sentence at the beginning of Line 142, explaining that the same mutations (Q65I and A86G) were introduced also in NascB.

Comment 7: Figure 4D-4E. Axis labels are missing.

Response: We have corrected it.

Comment 8: The CC1/2 should be added to Supplementary Table 4.

Response: We have added the CC1/2 to the new Supplementary Table 5 (original Supplementary Table 4).

Reviewer #3:

Comment 1: The manuscript is clearly written but could be perhaps a bit more concise. It would be useful to the reader to formally identify and name the different cytochrome P450 using CYP nomenclature (see <https://uthsc.edu/medicine/molecular-sciences/faculty-directory/nelson.php>) as the current nomenclatures can be a little confusing.

Response: Thank you very much for the good suggestion. It would be definitely helpful and facilitate communication if naming cytochrome P450 is following standard CYP nomenclature. However, in our previous study, we have already named the first P450 enzyme as NascB, and other groups also

named their discovered P450 enzymes in a similar manner. Thus, in order to unify the names of these enzymes that could produce NASs, we prefer to name them as Nas_{S1868} and Nas_{F5053} but with their CYP nomenclatures in the parentheses.

Comment 2: Could the authors clarify the difference between the strains of *Streptomyces* S1868 used in this study and *Streptomyces* NRRL S-1868 used in the study of Li and co-workers and their respective proteins NasS1868 and NasB, are they actually distinct?

Response: We are sorry for the typos. The strain of *Streptomyces* S1868 and *Streptomyces* NRRL S-1868 are indeed the same. We have changed *Streptomyces* S1868 and *Streptomyces* F5053 into *Streptomyces* NRRL-1868 and NRRL-5053, respectively. Comparing the sequences of P450s published by Li and co-workers with the ones in our study, we observed that Nas_{S1868} is identical with AspB, but Nas_B shared 98.8% sequence identity with NasB.

Comment 3: The presence of two substrate molecules in the active site in differing conformations is not unprecedented. However, it would have been nice to see some solution data alongside this study to help confirm whether this is a functional or dynamic aspect or merely a result of soaking excess substrate into a pre-formed crystal and as such an artefact of the crystallography. The use of UV-vis binding titrations and/or ITC data would have shown if there was any cooperativity during substrate binding as well as give binding affinities for each enzyme variant.

Response: We thank the reviewer for this valuable suggestion. As per the reviewer's suggestion, we used UV-visible spectrophotometer experiments to probe the interaction in solution between cW_L-P_L and each of three enzyme variants, i.e. Nas_{F5053}, Nas_{F5053}-Q65I-A86G, and Nas_{F5053}-S284A-V288A. Binding of cW_L-P_L to each Nas_{F5053} were all shifting a major Soret band from 418 nm to 387 nm, indicating transition of the heme iron from the low spin to high spin state. Plotting these spectral variations against cW_L-P_L concentration yields a

binding constant of $11.6 \pm 2.1 \mu\text{M}$ for the interaction between $cW_L\text{-}P_L$ and wild-type Nas_{F5053} , $25.6 \pm 1.0 \mu\text{M}$ for $cW_L\text{-}P_L$ with $\text{Nas}_{F5053}\text{-Q65I-A86G}$ and $4.81 \pm 0.26 \mu\text{M}$ for $cW_L\text{-}P_L$ with $\text{Nas}_{F5053}\text{-S284A-V288A}$ (Supplementary Figure 13). Our analysis result, that spectra titration data of Nas_{F5053} with $cW_L\text{-}P_L$ can be fitted to a rectangular hyperbolic curve, suggests that we could not observe any detectable cooperativity for substrate $cW_L\text{-}P_L$ binding on Nas_{F5053} . We have included those descriptions in a new section (Line 276-293) of the Results.

While we are revising our manuscript, the structural and functional characterization of NzeB (the same P450 as Nas_{F5053}) were published online at (ref17: <https://pubs.acs.org/doi/abs/10.1021/jacs.0c06312>). In their work, the mixture of $cW_L\text{-}W_L$ with $cW_L\text{-}P_L$ suppresses the homodimeric production formation from $cW_L\text{-}P_L$ and generates a heterodimer. In the structure of NzeB in complex with $cW_L\text{-}W_L$ with $cW_L\text{-}P_L$, $cW_L\text{-}W_L$ binds in the cyclization site ($cW_L\text{-}P_L\text{-}U$ site in our terminology) while $cW_L\text{-}P_L$ binds in the dimerization site ($cW_L\text{-}P_L\text{-}E$ site in our terminology). Collectively, their data also support that each substrate binds selectively rather than cooperatively.

In addition, we have added extra sentences from Line 488-491 to supplement that we have done both soaking and co-crystallization to obtain the complex structure. The identity of both complex structures serves to rule out the possibility of an artefact of the crystallography from soaking excess substrates.

Minor edits:

Comment 4: Line 124: add “in the” so reads “mutations in the N-terminal and C-terminal portions of Nas_{F5053} and NascB ”.

Response: we have added it.

Comment 5: Line 303: remove full stop following C2-N10

Response: we have corrected it.

Comment 6: Line 420: reword or remove “future” overexpressed” this terminology is confusing

Response: we have removed it.

Comment 7: Line 423: add concentration of FdR

Response: we have added it.

Comment 8: Figure 4: Labels on axes would be useful

Response: we have added the axis labels in Figure 4.

Comment 9: SI line 228: remove “of the” so reads “on Luria-Bertani (LB) media agar plates or in Lemoco liquid media”

Response: we have corrected it.

Reviewer #1 (Remarks to the Author)

This reviewer was convinced the changes in the revised manuscript.

As such, the paper is recommended for publication in Nature communications as an Article.

Reviewer #2 (Remarks to the Author):

The authors have fully addressed the concerns I raised in my assessment.